# On the Perfect Sphere: The Preference for Circular Compositions for Depicting the Universe in Medieval and Early Modern Art

Roger Ferrer-Ventosa

Department of Art History, University of Barcelona, 08001 Barcelona, Spain; roger.ferrer@ub.com

**Abstract:** This essay explores circular compositions in medieval and early modern art. Delving into the intersection of religious, philosophical, and scientific ideas, the text examines the prevalence of circular depictions in medieval and early modern aesthetics. Utilizing an interdisciplinary approach, the author draws from primary Hermetic and Neoplatonic sources, providing four reasons for this preference. Firstly, this essay explores the scientific understanding of the shape of the universe, planets, and stars. The second reason delves into the psychological, symbolic, and geometric aspects associated with circular compositions, connecting them to Christian cosmological diagrams and symbolism in the visual arts. Furthermore, the essay investigates the conceptualisation of the universe as a mirror reflecting the divine, emphasising the role of beauty in religious art. The essay concludes by examining the visual culture of medieval and early modern periods, tracing the evolution of circular representations from Roman coins and shields to illuminated manuscripts and paintings. The article sheds light on a hitherto underexplored aspect of medieval and early modern cultures, despite its significance in shaping symbolism and organizing iconographic programs within these periods.

**Keywords:** Hermetism in arts; Neoplatonism in arts; Christian aesthetics; circular composition; scientific cosmological diagrams; medieval visual cultures; early modern visual cultures; symbolic forms; *Lapidario*

> I behold our very beautiful world,
> united in a wonderful order (Nicholas of Cusa 1998, p. 37).

## 1. Introduction

How does a culture envision something as inherently ineffable, or at least beyond the human capacity for visualisation, as the entire universe and beyond? This essay will delve into an aspect of medieval aesthetics that linked religious, philosophical, and scientific ideas about the shape of the universe as well as ways of relating to it and representing it using circular compositions (Figure 1). For centuries, circular compositions depicted several spheres, combining Platonic approaches with Neoplatonic ideas, and Christian dogmas with scientific hypotheses, in an intuitive whole replete with different concepts.

One of the most compelling reasons behind this penchant for circular compositions has to do with the way in which the ancient Babylonian, Greek, Jewish, and Roman cultures conceived the universe as a circular entity, forging a vision that would subsequently give rise to the Christian worldview. Bianca Kühnel explores the contributions of the Babylonians, Egyptians, Ancient Greeks, and Romans in establishing general conceptions about the universe in Late Antiquity. Subsequently, Christian authors such as Isidore of Seville (c. 560–636) and the Venerable Bede (c. 672–735) expanded upon these ideas, shaping the prevailing way of conceiving the universe in the Middle Ages (Kühnel 2003, pp. 83–113). These foundational concepts were as pervasive and implicit in their time as they are in ours. In the twenty-first century, we have developed our understanding and imagination of the universe, influenced by figures such as Copernicus (1473–1543), Kepler (1571–1630), and Einstein (1879–1955), even without the necessity of directly reading their works.

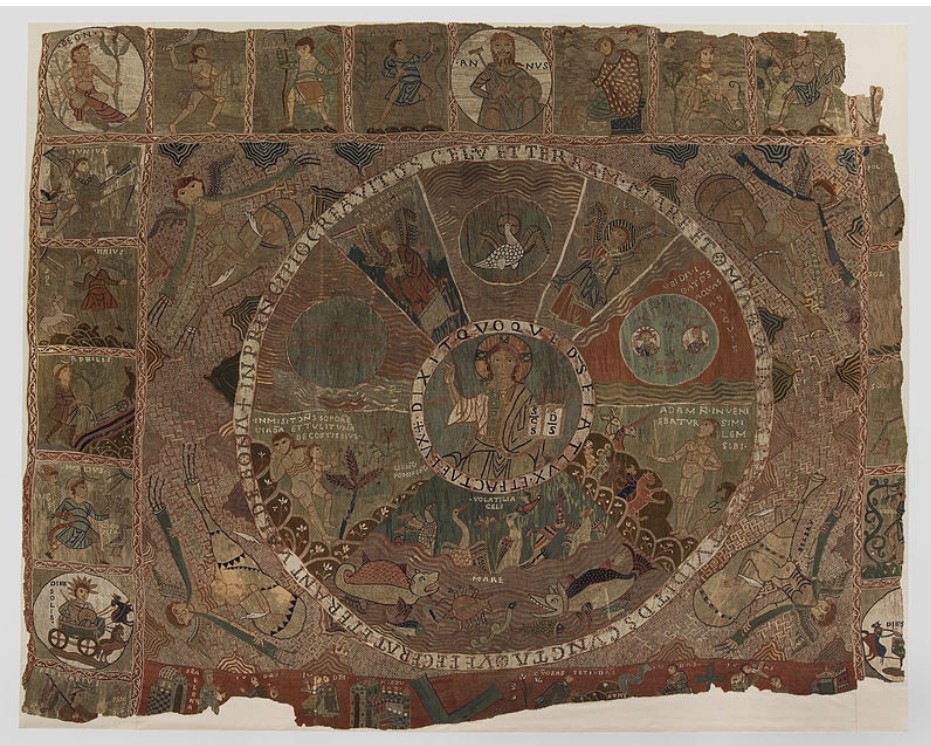

**Figure 1.** Anonymous, *Tapestry of Creation*, eleventh century. Embroidery, 3.58 m × 4.50 m. Girona Cathedral.

Accordingly, this paper performs an in-depth inquiry into the Hermetic and Neoplatonic cultural horizons that contributed to the development of the ideal of the perfection of the cosmos. In referring to Hermetism, the text alludes to Late Antiquity sources attributed to Hermes, and their influence on Christian authors of the Middle Ages, as well as, most importantly, early modern authors. These writings are closely associated with astrology, magic, and alchemy. Concerning Neoplatonism, it will refer to authors who drew ideas from Plato's *Symposium* (c. 370 BC), *Phaedrus* (c. 370 BC), and *Timaeus* (c. 360 BC), blending them with mysticism. Both philosophical and religious currents converged in early modern authors, proving highly influential in understanding the mindset of that epoch. Throughout the text, authors closely or loosely connected to this Hermetic–Neoplatonic cultural horizon will be cited, such as Plotinus (205–270), Proclus (412–485), Macrobius (c. 380–c. 430), Pseudo-Dionysius (sixth century), the *Corpus Hermeticum* (third century), the *Book of the Twenty-Four Philosophers* (twelfth century), Marsilio Ficino (1433–1499), Giordano Bruno (1548–1600), or Nicholas of Cusa (1401–1464).

This quintessential worldview was not only important in Late Antiquity and essential in the Middle Ages, but also prevailed in Renaissance and early modern aesthetics. As discussed further on, some of that current's theories and assumptions would serve as an aesthetic basis for the images created. Bianca Kühnel highlighted the prevalence of this geometrical shape in diagrams, particularly in the Byzantine context, and in the Western Carolingian and High Middle Ages periods (Kühnel 2003, p. 116).

As we explore in the following pages, this characteristic persists into the Late Middle Ages and early modern societies. This is one aspect in which we can observe a continuity between both cultural horizons. Surprisingly, there is a limited number of essays on this subject. Kühnel regrets that there are only a few cases where scientific diagrams have been integrated into art historical discourse (Kühnel 2003, pp. 66–67). This essay tries to contribute to filling this gap.

And, in addition to the primary Hermetic and Neoplatonic sources, this interdisciplinary essay will attempt to meet Kühnel's demands by analysing medieval circular compositions through art historical methodologies. It will also cite secondary sources in this

field of knowledge, such as Kühnel, Michael Camille, Rosa Alcoy, Alejandro García Avilés, and Linda Safran, as well as visual studies of the medieval period and its iconographic programs by Hans Belting and Ana Domínguez Rodríguez. But the essay will also delve into other fields of knowledge to analyse the issue, such as psychology applied to the perception of forms as Rudolf Arnheim did, or psychology in general with C. G. Jung. It will draw from secondary sources on Hermeticism from different schools of interpretation, Frances Yates and Wouter Hanegraaff, historians investigating the medieval cultural context closest to magical thinking, such as Paola Zambelli and Nicolas Weill-Parot, and cultural analysts like John Higgs.

In the subsequent pages, four reasons explaining this preference will be presented. Perhaps the more significant reason was scientific, and we will start from this point. Not only was the universe conceived as being circular, but so were the stars, planets, and movements of the astral forces. The emphasis is also placed on the importance of circular compositions in representing the universe and their role as imago mundi, namely, microcosms of the macrocosm. These imagistic diagrams were a means to represent and materialise the order of the cosmos, illustrating the harmonious relationships among its constituents (Safran 2022, p. 93).

The next topic addressed below is the reasoning behind the use of circular compositions, which is grounded in the analogous idea that nature is a mirror reflecting the divine. God, as the epitome of perfection, was often conceptualised or visualised in a rounded form, as exemplified in the *Omne bonum* (c. 1360). According to the theories of Plato (c. 425–348 BC), Nicholas of Cusa, and Giordano Bruno, visible things as a whole served as sensible images of the intelligible God.

In this vein, consideration is given to the notion that nature functions as a living mirror of God, the infinite deity, in addition to several other interpretations. Consequently, there was a conditional acceptance of matter in that it possessed the potential to unveil the divine. In the school of thought influenced by Hermeticism and Neoplatonism, beauty played a significant role in religious art. For instance, Bruno asserted that the visible world was akin to shadows or reflections of the One, and that it was possible to perceive this through imagination. Therefore, so as to understand divine ontology, it was first necessary to perceive its reflection in the mirror of nature. Art, particularly in depictions of religious figures like the Virgin Mary, became an essential visual tool for embodying beauty in the earthly realm.

In this study of circular compositions in medieval and early modern art, the spotlight is also placed on the symbolic and psychological reasons behind their popularity. The discourse extends to the dynamic interplay and tension created by the circular compositions within their rectangular settings, often found in walls, book pages, and similar contexts. Here, rounded curves and straight lines embody two contrasting forces, creating visual tension. Circular compositions possess the capacity to convey the notion of an outwardly expanding force, while simultaneously eliciting a sense of security.

Following this, the connection between geometry, mathematics, and mysticism will be examined, exploring the influence of the Pythagorean roots of Platonism when these artists or commissioners conceived the geometry of the piece. The symbolism and psychology associated with these compositions are then explored in various cultural contexts, including Christian cosmological diagrams.

Architecture, such as the combination of the square and hemispherical vault, is also discussed in relation to the symbolic representation of the celestial over the earthly. For similar psychological and symbolic reasons, this combination of cubes and spheres located in religious architecture is also to be found in some of the works of the great masters of the High Renaissance and Mannerism, such as Raphael (1483–1520) and El Greco (1541–1614), with the same use of the horizontal earthly dimension crowned by a circle as a visual expression of the divine realm.

Lastly, the preference for circular compositions is discussed on the basis of the tradition of composition deriving from Roman and even ancient Greek culture, the relevance of

the visual culture of previous ages, and their production of images. The origin of those images can presumably be traced back to the circulation of ubiquitous objects, such as coins, in many European kingdoms. An example of the importance of this visual culture is the constellations inscribed on medallions appearing in the *Lapidario* (c. 1250) and other illuminated manuscripts on astrology.

Regarded as a vade mecum or pharmacopeia, a handbook containing descriptions of medicinal substances, the *Lapidario* is devoted to astrological and talismanic themes, combining Western and Eastern influences. Specifically, it correlates stones with the degrees of the zodiac and decans by means of explanatory texts and illuminated miniatures. The book also sets out syncretic astrological theories with Neoplatonic and Ptolemaic influences, reflecting the interest of monarchs in astrology and astronomy[1]. Indeed, astrology played a central role in the courts of Alfonso X of Castile (1252–1284) and Peter IV of Aragon (1336–1387), among others, where Arabic texts were translated, and astronomical tables were drawn up.

Possibly originating from Roman medallions and coins, as well as from medieval cultures, these circular compositions eventually became symbolic and iconographic representations. For this reason, an analysis is performed here on the visual culture of the Middle Ages, above all from the thirteenth century onwards, and on that of the early modern period, focusing on the images appearing in illuminated books, tapestries, and paintings, and borrowing ideas not only from visual culture studies and the history of art, but also from philosophy, religious studies, the history of ideas, and the history of science.

## 2. The Shape of the Universe

The first reason behind the preference for circular compositions addressed here is the most obvious of all: to represent visually the purported shape of the universe. This representation was subsequently refined and nuanced over the centuries in Babylonian, ancient Greek, and Roman cultures, before being adopted by Christian culture in the Middle Ages and, albeit still rooted in previous ethe, adapted to the new worldview. In the *Catalan Atlas* (1370–1380), the known world up to that point is represented. In the section with cosmography (Figure 2), a model is included that was repeated well into the 17th century, with concentric circles from the Earth to the four elements, the seven "planets" (including among them the Moon and the Sun), the wheel of the zodiac, plus the phases of the Moon, along with a multitude of other astronomical data.

Special attention is paid here to the Platonic roots of this notion because of the importance of this current for the object of study, for the *Timaeus* already refers to a spherical universe as the appropriate shape for a living entity containing all living beings and, therefore, all shapes[2]. Indeed, Plato, for whom the sphere was the perfect geometric shape, visualized a universe formed by eight spheres[3]. More specifically, he used a dodecahedron as the quintessence to describe the universe, in reference to the circle; as it was a perfect living entity, it had to be circular in shape.

This idea was subsequently taken up by Neoplatonic thinkers, like Proclus, who had the following to say about the circle: 'The first and simplest and most perfect of the figures [. . .] If you divide the universe into the heavens and the world of generation, you will assign the circular form to the heavens and the straight line to the world of generation'[4]. Curiously enough, this Platonic and Neoplatonic stance became a model for visual compositions, as in Giovanni di Paolo's *Creation of the World and Expulsion from Paradise* (1445), the panel being clearly divided into two sections. On the one hand, it depicts the universe as a set of concentric circles, with the Earth at the centre surrounded by the heavenly spheres, including the Sun, followed by the constellations of the zodiac, above which God the Father, supported by blue cherubim, is floating. On the other, there is a representation of the world after the Fall, a realm of straight lines, curves, and the many chaotic forms of this protean reality.

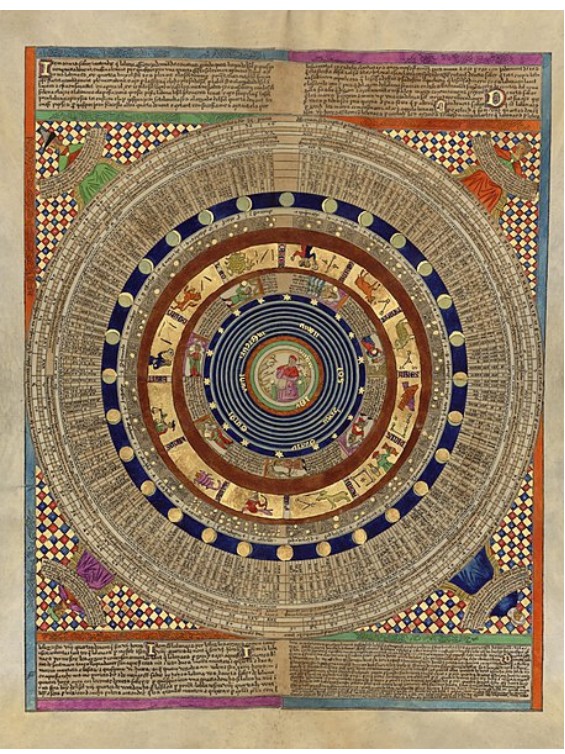

**Figure 2.** Abraham Cresques (?), *Atlas of Marine Cards* (*Catalan Atlas*), 1370–1380. Illuminated manuscript (parchment), 64 cm × 50 cm. National Library of France.

According to this worldview, all the extremes were located at the same distance from the centre[5], an idea that probably led Neoplatonic and Hermetic authors to formulate the theory set out in Aphorism 2 of the *Book of the Twenty-Four Philosophers*: '*Deus est sphaera infinita cuius centrum est ubique, circumferentia nusquam*' ('God is an infinite sphere whose centre is everywhere and whose circumference is nowhere')[6]. Both Cusa and Bruno probably borrowed this famous aphorism from the *Book of the Twenty-Four philosophers*[7], as it already appears in works written as early as in the twelfth century.

Those astronomers influenced by Pythagoras (c. 570 BC–c. 495 BC) and Plato looked for perfect movements when analysing the celestial model, the most perfect of which was circular, a notion also appearing in Aristotle (384 BC–322 BC) (*On the Heavens*, a reference work for centuries). Ptolemy (c. 100–c. 170) adapted some of Plato's and Aristotle's ideas, his model of the universe maintaining the movement of the spheres, but this now being eccentric. The poet Aratus (c. 310 BC–240 BC) disseminated the model among a wider audience, his verses—like, for instance, *Harmonia macrocomica*—becoming a celestial planisphere even down to modern times and his model being, more often than not, transformed into a visual scheme.

The description of the universe as a sphere formed by several spheres, a notion that prevailed for centuries, also appears in *Book II* of Pliny the Elder's *Naturalis Historia*, devoted to cosmology (c. 23–79). There is also reference to the motion of the spheres in the *Corpus hermeticum*, the most relevant Hermetic work. In his vision, Hermes sees the kyklophoria, which are explained to Asclepius as the circular motion of the heavenly spheres on incorporeal God, in opposite directions that create friction and balance[8].

Christian thinkers also pondered on this subject. For his part, the medieval scientist Robert Grosseteste (c. 1170–1253) argued that the spherical quality of the universe was a light effect: God inserted a point of light into a point of matter, the former radiating in all directions pulling the latter spherically[9]. As a consequence of this process, the density gradually increased from the circumference to the centre of the Earth (Quinlan-McGrath 2013, p. 26). When artists wanted to represent God's creation, they worked with these ideas in mind, following the biblical verse appearing in one of the

Sapiential Books, the *Book of Proverbs* 8, 27: '[...] when he marked out the horizon on the face of the deep'. God traces a circle over the abysm as the first act of creation, as illustrated in an outstanding folio from the *Codex Vindobonensis 2554* (c. 1225) (Figure 3).

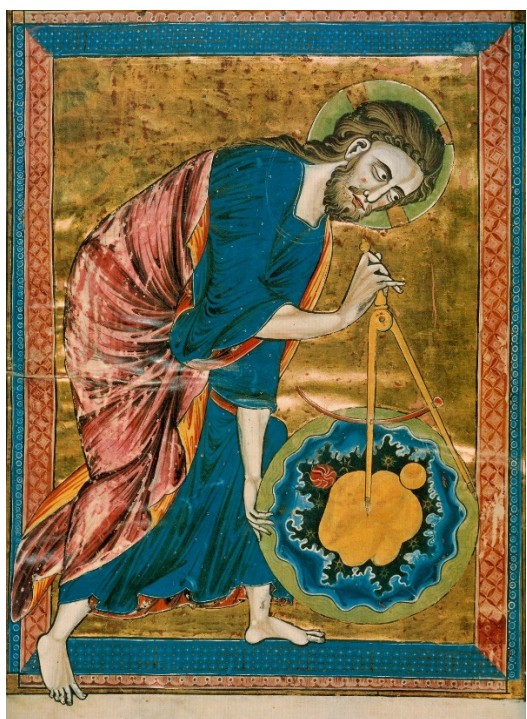

**Figure 3.** Anonymous, *Codex Vindobonensis 2554*, c. 1225. Illuminated manuscript (parchment), 34.4 cm × 26 cm. Österreichische Nationalbibliothek, n. 2554, f.1 v.

In Christianity, with the circle thus came the representation of the compass as an attribute of God the Creator. This can be seen in another example appearing in a section of the Catalan artist Pere Serra's *Retable of the Holy Spirit* (1394), in which God is depicted creating a spherical universe, an idea that would subsequently become a subject of theological reflection[10].

Furthermore, Christian thinkers and artists gave God a leading role in their depictions of the universe, adding an Empyrean conception beyond the physical spheres, with the Christian God and angels, as is the case of several medieval and early modern images. This model with the Empyrean surrounding the spheres is shown in one of the coloured copies of the *Nuremberg Chronicle* (1493) (Figure 4).

Not only the physical universe but also the astral subject-objects and their movements were thought to be circular. The universe was spatially conceived as a sphere formed by several other spheres moving in a circular orbit, as well as being established as a circle, like the zodiac wheel; the same also applied to the constellations depicted in circular compositions[11]. Even time was believed to be circular, taking the shape of a cosmic snake or one eating its own tail, to wit, an ouroboros or the image of the cycle of time.

This cosmological conception originated before the ancient Greeks, for the Sumerians and the Egyptians shared some of those ideas. Albeit generally not depicted on their vaults, which were rectangular, the Egyptians also conceived the sky as a circle, as a figure of the goddess Nut whose body was adorned with stars and solar discs. Surrounding her, the Egyptians placed the so-called 'Nun', the last sphere which was beyond the light and even unknown to the gods (Lull 2016, pp. 392–93). According to Lull, however, the Denderah zodiac, perhaps the most famous ancient Egyptian image of the cosmos, is the only circular composition of a celestial planisphere, dating as late as the first century BC, in that culture (Lull 2016, p. 413).

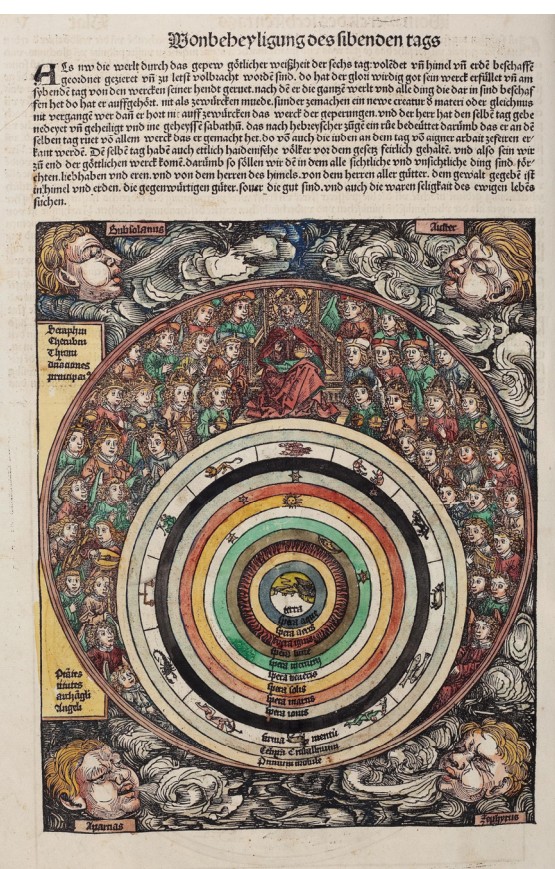

**Figure 4.** H. Schedel, *Liber chronicarum* (*Nuremberg Chronicle*), 1493. Printed (incunable), 44 cm × 30 cm. University of Heidelberg. https://doi.org/10.11588/diglit.8305#0032 (accessed on 1 October 2023).

Those ideas gave rise to an iconographical type: the circular zodiac[12]. In the *sphaera barbarica* and *sphaera graecanica* alike, draftsmen included allegorical representations of the signs of the zodiac and the constellations, which were conceived as spheres.

The way of representing the zodiac wheel was quite firmly established. The *Revised Aratus latinus*, appearing in an illuminated book (Figure 5) housed in the Abbey Library of St Gall (Ms. Sang. 250) (fourth quarter of the ninth century), includes a common illustration of the zodiac wheel—depicting Apollo-Helios and Diana-Selene in the centre—with the signs proceeding in counterclockwise direction. The order of the constellations on the ecliptic had a direct influence on the zodiac and, subsequently, on natal charts, with the 12 houses, beginning with the ascendent on the left side. Around that time there were representations with the same pagan iconographic type appearing, for instance, in Rabanus Maurus's *De originibus rerum* (c. 825).

With some modifications, it follows a model already present in Late Roman, Early Byzantine pagan, and Jewish works of that period. The god in the central disc dominates the zodiacal signs, symbolizing both nature and time, with the sun more prominent than the moon, adding a higher status to this figure (Kühnel 2003, pp. 45, 164)[13].

One of the most noteworthy representations of the cosmos as a sphere is the *Farnese Atlas*, a Roman marble copy of a probably Hellenistic original, now lost, specifically a celestial globe with low reliefs depicting most of the ancient Greek constellations. Another more complex model associated with the zodiac wheel is the *paranatellonta*—*para* in Greek means 'along with' and *anatellein*, 'to rise'—each of whose 36 figures occupies 10 degrees, albeit sometimes containing double the number, with each of the 72 figures consequently occupying 5 degrees (Domínguez Rodríguez 2007, p. 309)[14]. These depictions of the celestial sphere and the different constellations were used for mnemonic purposes, as a visual scheme for remembering the stellar positions and as a useful scientific method.

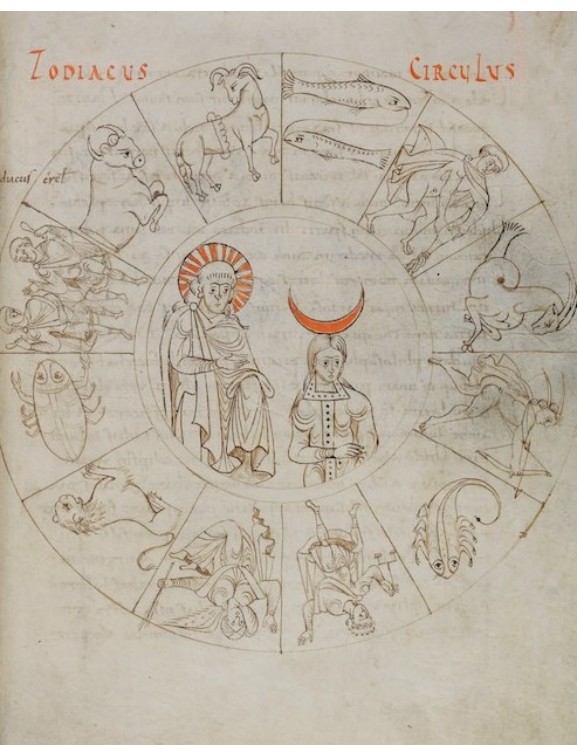

**Figure 5.** Anonymous, *Zodiacus Circulus-Sol & Luna*, in 'Revised Aratus latinus', fourth quarter of the ninth century. Illuminated manuscript (parchment), 24.7 cm × 18 cm. St Gallen.

Those astronomical images were important because they were givers of energy linked to the sacred which, in turn, was believed to be a perfect and immutable order, sanctioned by tradition, with the same stable formulas as icons. Furthermore, humans could benefit from pondering on them: '*Certains hommes, grâce à leur compréhension de la disposition céleste, comprirent beaucoup de choses cachées dans le monde des éléments, en explorant les secrets de la nature aussi bien supérieure qu'inférieure*'[15], as will be seen further on.

For example, humans could realize that they were a microcosm of the macrocosm (See Ferrer-Ventosa 2023, pp. 495–511), whose goal, as Quinlan-McGrath asserts, was to 'look outward, and see the divine order; look inward, and recognize the divine soul' (Quinlan-McGrath 2013, p. 198)[16]. That notion, based on an analogy, which is a constant in both Hermetic and Neoplatonic literature, could be depicted in a circular diagram with an anthropomorphic figure in the centre, surrounded by the spheres of the elements, planets, and signs, drawing parallels between these and human body parts and character traits (Page 2002, pp. 52–54). From a Neoplatonic viewpoint, the visible image should link the mind of the believer to the invisible archetype, an idea that was acceptable in Christian terms[17].

Even Dante (c. 1265–1321) held that view of the universe, for he structured Paradise in his *Divine Comedy* (1321) along the lines of the Ptolemaic model of nesting spheres, which ends in a vision of God: three circles of different colours but of the same size merge in pure light; then, the poet spies a human figure, a reference to the second person of the Trinity ('*mi parve pinta della nostra effige*') (Dante 1997, p. 458). It is a vision of the One replete with Neoplatonic ideas, exemplified in the last verse, with love at the centre of the cosmos moving the stars.

In the fifteenth and sixteenth centuries, philosophers with Hermetic and Neoplatonic worldviews maintained some of these principles. For instance, in his *De vita* (1489), Marsilio Ficino describes the celestial revolutions and circular movements as focal points[18]. As those astronomical images accumulated spiritual power, he contended that it was essential to live with them. In his theory of images, the material image of each object and being spread its ra-

diation spherically from every point, losing its original shape in this radial propagation; that radiation carried the form and matter of the material image (Quinlan-McGrath 2013, p. 76).

In order to attract the universal rays for the purpose of promoting good health, those images were depicted by tracing the movements of the planets and their passage through the signs of the zodiac in paintings with magical attributes. Ficino recommended that they be painted in bedrooms, where they were conducive to sleep and meditation with the aim of attracting cosmic energies, like the virtues associated with the planets[19].

And in Ficino, as in other Hermetic authors, the way of considering the universe was also influenced by Muslim authors with Neoplatonic and Hermetic leanings, such as Abu Ma'shar (787–886) and the anonymous author of the *Picatrix* (c. 1050), who used the system of Teucer of Babylon (first century BC), the *sphaera barbarica* that combined notions deriving form Hellenism, Egyptian astrology, and Babylonian culture, instead of the Ptolemaic system, known as the *sphaera graecanica* (Domínguez Rodríguez 2007, p. 336). In both cases, the zodiac system was conceived as a circular structure, specifically, as a sphere.

Bruno, a philosopher who borrowed some of his ideas from Hermetism, put forward his own hypothesis based on spiral motions. Despite praising Copernicus, his system was original, drawing from the premise that as celestial bodies were formed by combinations of different elements, their motion could not be simple or uniform. Bruno highlighted the distance between an infinite universe and the supposedly geometric harmony of a spherical cosmos. To Tessicini's mind, Bruno's model was geometrically inaccurate, for it contradicted both Copernicus and traditional astronomy (Tessicini 2016, pp. 156–57). Bruno applied some of the Neoplatonic and Pythagorean principles of harmony of the universe, such as the cosmological order, to which he added his own concept of the infinite that transformed spheres into spirals.

This worldview changed with Kepler's innovative way of conceiving the cosmic order which, nonetheless, was grounded in Neoplatonic thought. In his *Mysterium Cosmographicum* (1596), he declares that the sphere is perfection and that, therefore, the planets have to be spherical.

### 3. Nature as a Mirror of God

The second reason is more metaphorical: if God is circular owing to His perfection, then nature, which mirrors Him, must also have this shape.

A miniature illustration in the *Omne bonum* is divided vertically into three sections, in the lowest of which there is a vision of the cosmic diagram with two donors, a layman and woman, contemplating it with expressions of devotion (Figure 6). They are praying for their souls in the physical realm, the sublunar world under the spheres. In the middle section, St Paul and St Benedict are interceding for them. Lastly, in the top section, there is a vision of God with a round face to highlight visually His perfection, as is contended here. The divine face of God shines for the angels, the saints, the donors of the piece, and the beholder (Camille 1996, p. 126; Alcoy 2017, pp. 91–92). As God has a perfect form, He has to be conceived as being circular. According to Cusa in his *De ludo globi* (1463), the true circle only exists as an ideal, for any material thing that appears to be truly round is not really so, for only God can possess perfect roundness[20].

As to its structure, however, the illustration appearing in the *Omne bonum* points to a relationship between invisible God and the world, underscoring the similarity between the shape of both so as to explain visually a principle of Christian theology[21]. As nature, or the world, is a work of God, it is more excellent that any artwork created by humankind, as St Bonaventure claimed (Camille 1996, p. 134).

One of the reasons why these circular forms with that meaning were used was linked to Hermetism and Neoplatonism: the consideration of nature as a mirror of God and the artwork as a reflection of that intuition. This notion that the visible reveals the invisible appears in Plato's *Timaeus*, a work on cosmology and ontology and one of the seminal sources of these currents. It ends by expressing the idea set out here: the philosopher

declares that the visible things of this world are sensible images of the intelligible god ('image of the intelligible Living Thing')[22].

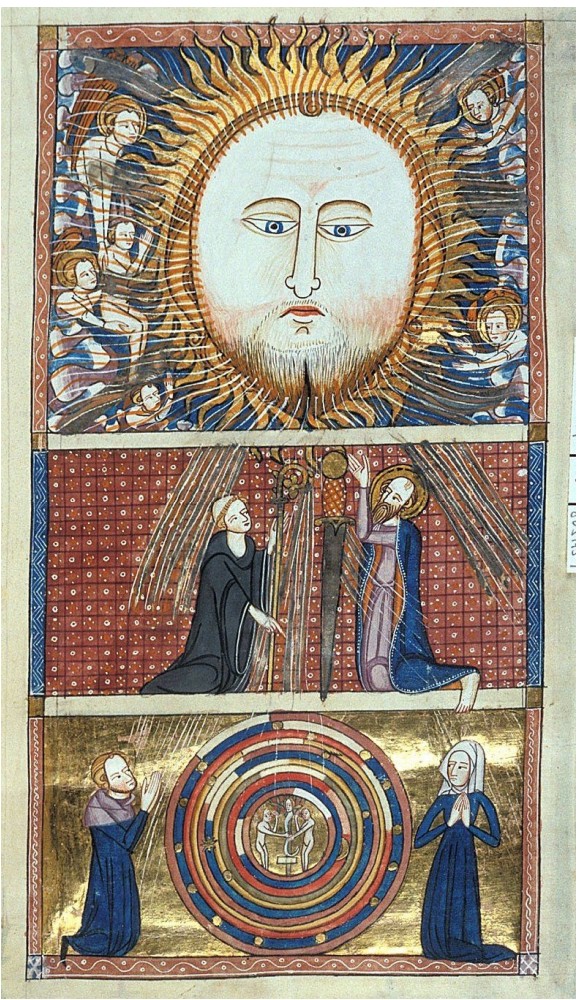

**Figure 6.** James le Palmer, *Omne bonum*, c. 1360. Illuminated manuscript (parchment), 45 cm × 31 cm. British Library, Royal MS 6 E VI/1, fol. 16r, http://www.bl.uk/manuscripts/Viewer.aspx?ref=royal_ms_6_e_vi!1_fs001r (accessed on 1 October 2023).

This thread can be followed in the philosophical current essential to the arguments set out here. The analogy between the intelligible and its reflection in the material world had a long tradition in the Platonic/Neoplatonic and Hermetic schools. Nature would be another reflection of the primal image. For most of these authors, nature was neither evil, as some Christian authors held, nor an impersonal machine, based on mechanisms, as in the subsequent Cartesian and Hobbesian mindsets, but the receptacle of God, in the sense of both the Neoplatonic conception of beauty and the character of Diotima in Plato's *Symposium*, who advocates for embodying beauty in this world (Hanegraaff 2022, pp. 106–12).

The notion of the things of the world as shadows or reflections in a mirror chimes perfectly with Neoplatonic ontological perspectives. As Geurt Imanse asserts, occultism is that in 'which we see the reflections of the visible world, it is the mirror of visions and the mechanism of magical life' (Imanse 1987, p. 357). Those things reflected in matter are like shadows in that they are the product of the light of the real being or the One, as well as being reflections of this One in matter, that is, not the real being but a product of the One, an image but not the real subject or model, an optical illusion, so to speak. The intellect only sees the appearance of things, which are signs. According to some medieval Neoplatonic authors, like Pseudo-Dionysius and Abbot Sugar (c. 1080–1151), divine splendour can be

perceived and grasped through images, *per speculum in aenigmata*. Scholastic thinkers drew a distinction between beauty and kindness, the latter being a moral attribute differing from sensible things (Camille 1995, p. 340).

The *Tapestry of Creation* (Figure 1) (eleventh century) in Girona (Spain), a kind of mandala imbued with Christian theology, is an encyclopaedia of the universe of sorts, but as a mirror image. From a Christian viewpoint, Christ is at the centre of the cosmos, from which the entire creation expands.

In his *Docta ignorantia* (1440), Cusa, who was influenced by that philosophical school, declares that nature serves to reveal the divine, the visible offering a faithful reflection of the invisible, in order that all creatures should understand the Creator through these mirror images and enigmas, literally stating, 'All our wisest and most divine teachers agree that visible things are truly images of invisible things and that from created things the Creator can be knowably seen as in a mirror and a symbolism'[23].

Bruno, who praised Cusa in his *Oratio valedictoria* (1588)[24], considered the order of nature as the living mirror of the infinite deity (Tessicini 2016, p. 132). The motto *per visibilia ad invisibilia Dei* corresponds to this mindset. In his *Thirty Statues* (1586), Bruno argues that human intelligence is finite, so the only way to understand divine ontology—ultimate infinity in the words of Cusa—is by seeing it reflected in the mirror of nature, in its figures, shadows of truth, effects, or traces[25].

The cosmos is 'a sort of living mirror in which is the image of the natural and the shadow of the divine'[26].The inner spirit would be like a mirror in which the perceptible object is at one with the percipient subject[27]. Imagination is essential for human intellectual processes and Bruno grounded his system in this ability of the mind, quoting the Neoplatonist Synesius (c. 373–c. 414): when awake, a human being can be wise, but when asleep, participates in God and His methods because of the power of imagination[28]. As Bruno declares further on in the same work, vision is more certain in the act of imagination than in the physical world[29], because the constitution of things, not just their appearance, is seen directly.

The philosopher from Nola divided beings into those that were things and those that were their signs. The cosmic order was rational, an image or vestige of the divine[30]. The best example of this way of thinking in Bruno, which considers things as shadows or vestiges, would be images as shadows of ideas, which was the title of one of his major works.

In another of his magna opera, *De gli eroici furori* (1585), Bruno suggests that the enlightened sage sees God reflected in the world and in nature, unity in numbers, and the Monad in plurality. Although this divine Monad remains in the realm of intellectual substances, the earthly and cosmic nature, its reflection, arises from it[31].

Hence the conditional acceptance of matter, for it could reveal the divine. The problem of the visibility of God raised a number of crucial questions about the relationship between visible things and matter in general in Christianity. As for the range of possibilities, at one end of the scale, it would imply regarding everything in matter as false, with truth—the only real aspect in ontological terms—being the sole divine essence, a stance that had serious visual consequences, such as aniconism and iconoclasm. Indeed, at the Synod of Elvira (c. 306), the Church forbade images in churches contending that the object of veneration and worship did not belong on a wall (Belting 1994, p. 145).

At the opposite end of the scale, there was the idea of spirit and matter as one, the latter with the ability of offering insights into the divine, a monist solution that supported Bruno's arguments. Ficino and other Christian thinkers with a Neoplatonic background took an approach that was not as drastic as Bruno's, according to which matter would hint at the divine. The utopian writer Tommaso Campanella (1568–1639), a thinker influenced by them, expressed this idea with a metaphor. In his *Epilogo Magno* (1598), he refers to the world as a statue of God, while stating that true philosophy involves searching for evidence of the godhead in nature, like a lover contemplating the portrait of the loved one[32].

In other words, spiritual beauty was transformed into material beauty, hence the controversial conception that the Virgin could not be ugly; as a matter of fact, some

theologians criticized icons and sculptures that were not sufficiently beautiful. Nevertheless, beauty was regarded as a quality, as evidenced by the story appearing in the *Stella Maris* (c. 1248) about a painter who painted a very beautiful Virgin and a hideous devil. Piqued, the devil demanded that the painter portray him as beautiful, as well. When the painter refused, the devil made him lose his footing on the scaffold. Nevertheless, the painter did not plunge to his death because the image of the Virgin, pleased with the beauty with which she had been represented, extended her pictorial arm and supported him until he received human assistance (García Avilés 2021, pp. 105–6).

Reality could be understood as a ladder with several rungs leading from the ground or matter to the higher or intelligible, an idea espoused by Renaissance Neoplatonists, such as Ficino in his *Theologia platonica* (1482)[33], although without literally mentioning the ladder. For Paola Zambelli, it was an idea very close to the heart of this philosophical current (Zambelli 2012, p. 107). Even a thinker influenced by Neoplatonism like Bruno delved into this notion. As with Ficino, in his *De la causa* (1584), he imagines a ladder uniting all levels of existence, namely, the Neoplatonic One. By climbing or descending it, the godhead was able to communicate with even the smallest thing. But even magi could reach the top of the ladder with their rites[34].

## 4. Circular Compositions and the Symbolism and Psychology Deriving from the Forms Depicted

Another reason behind the use of circular compositions for visually representing the universe had to do with their benefits. As previously seen, this conception of the universe as a sphere was portrayed by medieval and early modern artists but not solely for mirroring physical reality.

These circular compositions stand out for the fact that they practically always focus the attention of the beholder. Besides the two types into which Rudolf Arnheim divided these compositions in his *The Power of the Center*, concentric and eccentric, he includes another in which there is something that disturbs the circle's balance, like, for example, an off-centre ink dot (Arnheim 1982, p. 6). Arnheim is of the mind that circular compositions promote movement, for their centre points in all directions but to none in particular. So, despite having an indisputable centre, this and their circumference were all points but none, as in the quote from the *Book of Twenty-Four Philosophers* or in Bruno's conception of the infinite. An accomplished circular composition transmitted the feeling of an outwardly expanding force striving to manifest itself, stretching itself to surpass its limits (Arola 2015, p. 148). Perhaps the best example of this force is a star, like the Sun.

The power that this conception of the image confers on the centre is discernible in the models of the universe discussed above. Some examples, such as the one appearing in the *Leiden Aratea*, a famous Carolingian manuscript (815), are literally full of circular elements, including some anthropomorphised planets, constellations, and signs, inscribed in a circle.

In illuminated manuscripts, it is possible to observe the tension between two opposing forces, as the drawings have two limits: the rectangular shape of the folio and the circular shape of the image, in the case of sheets of paper, papyri, and scrolls.

Circular compositions are particularly limited when they are tondi, designed to be hung anywhere depending on the whim of their owners. As Arnheim indicates, a round frame, or tondo, contrasts with its surroundings by creating the feeling of isolation, whereas a rectangular frame usually adapts to its surroundings, since the walls on which it is generally hung are almost always rectangular (Arnheim 1982, pp. 50–54, 115–35). In relation to the Renaissance tondo, Bouleau claims, 'A tondo (circular picture) is nearly always a simple and charming picture, with supple forms enclosed in a ring; and at a time when a monotonous rigidity dominated the altarpieces, it offered from the very first its slightly precious grace, freed from symmetry' (Bouleau 2014, p. 49).

Raphael's masterpiece *Madonna della Sedia* (1514) shows how the shape of the frame determines the structure (Figure 7). As already noted, it gives the impression of centrifugal force, that the artist has composed the painting from the centre outwards. Raphael achieved

that feeling in circular compositions, according to Gilbert Durand, for whom they conjure up a feeling of safety, mystical enclosure, stability, and intimacy inside an initiatory labyrinth, as mandalas do (Durand 1999, pp. 238–41).

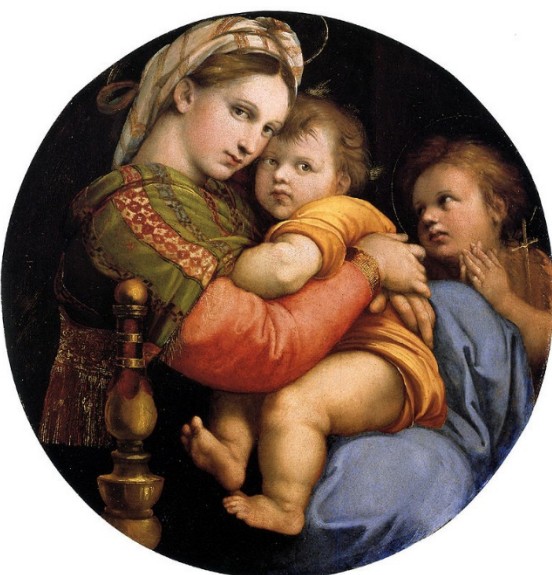

**Figure 7.** Raphael, *Madonna della Sedia*, 1514. Oil on panel, 71 cm × 71 cm. Palazzo Pitti (Florence).

Geometry and mathematics, from a mystical viewpoint, were keystones of these cultural horizons. In the Platonic and Hermetic context (or contexts), it was considered that each entity and species had a specific mathematical shape, which derived from a particular idea from above (Quinlan-McGrath 2013, pp. 147–48). As a rule, Neoplatonists believed that the universe possessed a mathematical structure. In addition, those involved in the arts believed that they should be able to mimic this structure[35].

In this process of transforming conventional symbols into divine ones, Cusa proposed a three-step system that allowed access to the infinite maximum, the divine in his terms, which, in his approach, was based on mathematical signs. The three steps consisted first of considering mathematical figures with their finite qualities and ratios; next, those ratios had to be transformed into infinite figures; and finally, the ratios of infinite figures had to be established on a higher, supreme plane, namely, the infinite[36].

Research on the golden ratio, deriving, among other things, from the circle[37], reveals this mystical approach to mathematics. Following its advent in Florence, Renaissance artists—including Uccello (1397–1475), Michelangelo (1475–1564), Piero (c. 1415–1492), Botticelli (c. 1445–1510), Leonardo (1452–1519), and Raphael, to name but a few—certainly used it.

But it was the author Luca Pacioli (c. 1445–1517) who presumably extended its use or, at least, who excelled in the art of constructing his hypotheses on solid mathematical foundations. His *De divina proportione* (1509) is an outstanding treatise replete with Neoplatonic concepts. He disseminated his own ideas, in addition to Plato's and Piero della Francesca's works on the golden ratio, written 40 years before.

In addition to addressing some of the principles of the golden ratio in his treatise, Pacioli was the first author to regard it as divine (Livio 2003, pp. 127–47), using the idea of correspondences between things with no apparent relationship[38], a key Hermetic concept, to support his arguments. In his *De divina proportione,* he demonstrated how, on the basis of this proportion, it was possible to obtain simple geometric shapes and, therefore, regular bodies, namely, with volume, and from these the rest of the figures.

More to the point, in Chapters LVI and LVII, Pacioli explains how to construct a spherical body and to insert it into the five Platonic solids (Pacioli 1956, pp. 107–8). For this reason, the Platonic quintessence was the pentagon, whereas for Pacioli, the dodecahedron

was the perfect figure, the embodiment of the circle, which he created using 12 pentagons[39]. This Platonic conception was embodied in paintings not only for its benefits, owing to the formal reasons already set out above, but also for the psychological factors underlying their symbolism.

As with others, medieval Christian cultures had codes to explain things visually. For instance, the circle corresponded to the celestial vault in the traditional symbolism of forms[40]. Similar to other cultural analysts, Roman Gubern maintains that the circle would have formed part of a universal symbol code, a figure of perfection since all the points on its circumference are equidistant from its centre, a symbol of both the Sun and the patriarchal chariot, as well as the maternal full Moon (Gubern 1996, p. 80). On his part, the specialist in symbols of Western Esotericism Frederick Goodman delves a little deeper in this respect, with the circle symbolizing the spirit and the cosmos as a whole. By his reckoning, this form is certainly the most important of all magical symbols (Goodman 1989, pp. 16–17)[41].

As another cultural analyst John Higgs remarks: 'We all need models in order to deal with the world around us. We need models that fit the existing facts, and which have some ability to predict what will happen next. This is what all the best ideologies, religions and philosophies offer us. What we shouldn't do is confuse these models with the real world, for the map is not the territory and the menu is not the meal' (Higgs 2013, p. 73).

Asian mandalas offer insights into this question. According to Jung, their shape is a psychic structure that exists a priori in the human mind and, therefore, an archetype inherent to the collective unconscious[42]. Buddhists and Tantrists use the mandala, a Sanskrit noun signifying 'circle', to symbolize the essence of a sacred space, a matrix or model of a perfect universe. These geometric shapes usually consist of an inner circle enclosing a god, this main deity being surrounded by other deities, bodhisattvas, buddhas, lamas, and so forth, each functioning as a specific power, and of a multilevel square palace inside a circle, again surrounded by figures (Leidy 1997, p. 17; Thurman 1997, p. 127). Indeed, there are examples of cosmological compositions with astrological connotations in the Buddhist mandala genre (Leidy and Thurman 1997, p. 75).

As with mandalas, Christian cosmological diagrams represented a journey, in this case, through the universe to the sublunary world. Plotinus had already explained creation as a journey to the centre, instead of outwards. In *The Sixth Ennead* (c. 270), he portrays the journey of the soul as a circle that does not lead to the outer reaches but to the centre, the point from which it emerged[43].

Similar types, like the remarkable Persian horoscope of Prince Iskandar (1384–1415), showing the planetary positions when he was born on 25 April 1384, are also to be found in the aniconic tradition of Muslim countries (Figure 8).

But as Mircea Eliade claims about Eastern cultures, these images are imago mundi, an image of the world or worldview, representing a thumbnail universe. When one is created, the creation of the world is being recreated magically (Eliade 1963, p. 24–26). In both Eastern and Western cultures, circular compositions and structures can be considered as imago mundi. If the universe is circular, with a limited form, it can become an equally limited image of the whole, an imago mundi that would represent a microcosm of the macrocosm[44].

The circle with psychological and spiritual connotations was put to several uses in Christianity, such as the mandorla enveloping Christ in majesty and the nimbus around the heads of saints, both variations on the same theme. The association of the clipeus, the circular mandorla, with heaven is present in many imago clipeata compositions. From Antiquity down to the Middle Ages, this was a way of making the invisible divinity visible (Alcoy 2017, pp. 145–47).

The same symbolic code can be applied to architecture. The relationship between matter and spirit, earth and heaven, human and divine, is symbolically portrayed in some buildings associated with the idea of divinity, including St. Peter's Basilica (plus the Pantheon in a pagan context), in the combination of the square and the hemispherical vault. In view of the spherical dome crowning the cube, it can be interpreted in the sense of the

celestial prevailing over the earthly. As the square was regarded as inferior to the circle, the latter was associated with the sky and the former with the earth[45].

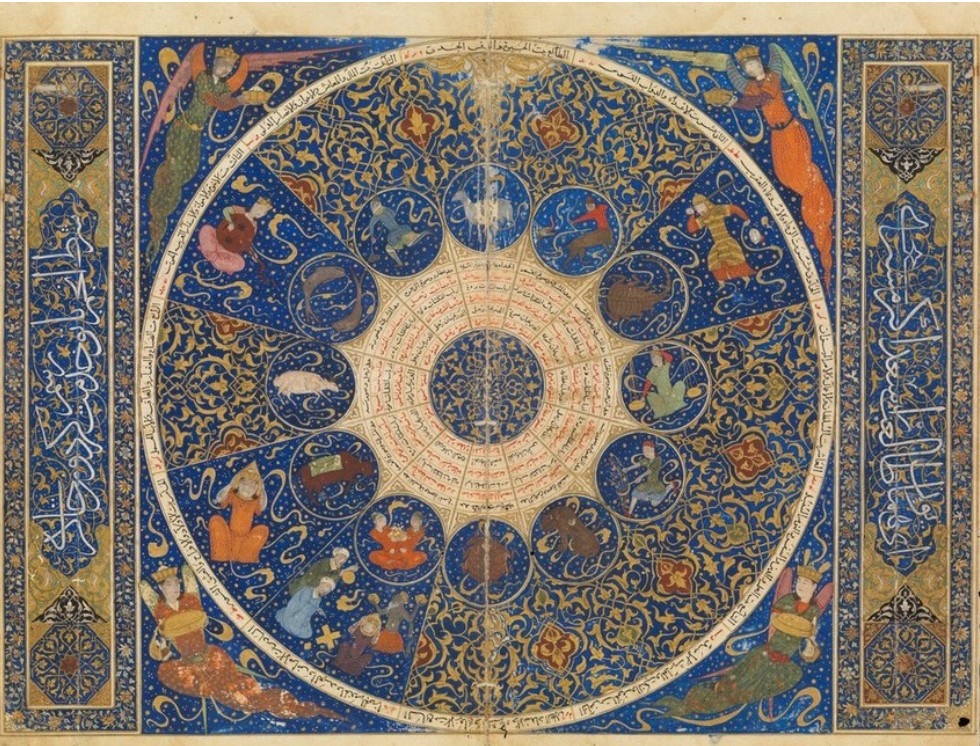

**Figure 8.** Anonymous, *Kitab-I viladat-i Iskandar (Book of Birth of Iskandar)*, 1411. Illuminated manuscript, 26.5 cm × 17 cm. Wellcome Collection, MS Persian 474, pp. 38–39, https://wellcomecollection.org/works/ua87equq/items?canvas=38 and https://wellcomecollection.org/works/ua87equq/items?canvas=39 (accessed on 1 October 2023).

The shape of semicircular apses and apsidioles, as well as the themed frescoes and mosaics which frequently depict the Heavenly Jerusalem, the Kingdom of Heaven, the Second Coming, and the like, contributes to reinforcing the link between God, perfection, and the circle.

Even the rose windows of churches were interpreted symbolically. For example, the different sections of the four levels of concentric circles of the Gothic north rose window at Notre-Dame, in Paris (thirteenth century), are separated according to the golden ratio (Meisner 2018, p. 111)[46]. That Romanesque and Gothic porticos had circular rose windows has been attributed to the compositional influence of cosmological designs appearing in illuminated manuscripts, often featuring a central Christ (Arnheim 1982, p. 131).

A traditional way of conceiving this Christian symbolism can be seen in an engraving from the alchemical treatise entitled *Tripus aureus, hoc est, tres tractatus chymici selectissimi* (Figure 9), whose 1677 edition includes an engraving featuring a radiant circle representing God, the circle as the world creation, made by Him, with angels of divine perfection, a triangle as the material world of the four elements (and the *Tria prima* of alchemy), and a rectangle below as the inferos, the world of evil, visually symbolised by a half sphere or Satan.

This Christian symbolism persisted in the visual arts during the Renaissance. Over the following centuries, symbolic values would still be attributed to the circle, as a figure linked to the divine and perfection. Something akin to this symbolism of the forms also appears in the paintings of some of the great masters of the period, such as Raphael and El Greco.

Raphael, who had a penchant for these simple forms, resorted to them in his *Disputation of the Holy Sacrament* (1510) (Figure 10). This composition plays with the contrast

between the horizontal line of the theologians and the curved line of the divine above them, specifically, the circle in which Christ is depicted, crowned by the Father, both in turn exerting an influence on those below. Furthermore, this same contrast can be found on the vertical axis between the rectangular altar and the series of circles and semicircles decorating it. But the real power of the composition resides in the two opposing forces: circular versus rectangular. The lower one depicts a number of saints, theologians, and pontiffs, arranged in a rectangular shape, resembling a frieze, a horizontal line that curves at its ends. Meanwhile, the lower edge of the upper level rests on a bank of clouds upon which saints and key figures from the Testaments are seated, surrounded by other beings from the Catholic otherworld, such as archangels, seraphim, and cherubim—all figures placed in concentric circles emanating from the axis of God.

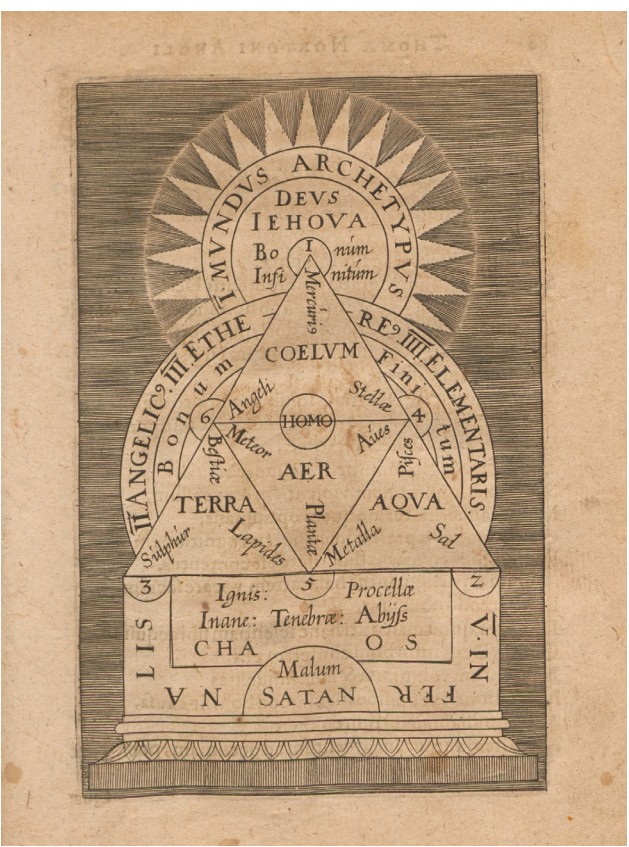

**Figure 9.** Thomas Norton, 'Crede mihi seu ordinale', in *Tripus aureus, hoc est, tres tractatus chymici selectissimi*, 1618. Engraving, 19 cm × 13 cm. ETH-Bibliothek Zürich, Rar 8239. https://doi.org/10.3931/e-rara-34295.

The central axis features the Trinity, following a typical iconographic scheme, with the Father at the top, followed by the Son and the Spirit, represented by the customary dove symbol. Lower down, there is a monstrance with the consecrated host, as the issue in dispute is the existence of transubstantiation. The painting defends its existence with stimulating visual arguments, placing the liturgical symbol representing it directly below the Trinity on the central axis. This arrangement serves to connect the divine circles with the rectangular forms of the earthly dimension, represented by Doctors of the Church, popes, and even writers such as Dante. Even the ground exhibits a design featuring squared shapes.

Raphael used that kind of composition in other masterpieces, such as the *Transfiguration* (1520). El Greco borrowed the compositional scheme of Raphael's *Transfiguration* in *The Burial of the Count of Orgaz* (1586), whose forms possess an analogous symbolic meaning, although he painted this masterpiece in the late Mannerist style characterized by its swirling

shapes. The divine characters are arranged in a circle around Christ on the throne in the upper segment, taking advantage of the fact that this portion is semicircular in shape. In contrast, the human figures, representing matter, are to be found in the lower square-shaped section. They include the quintessence of the material realm—a corpse—together with a frieze of pale faces that starkly contrast with their Baroque attire.

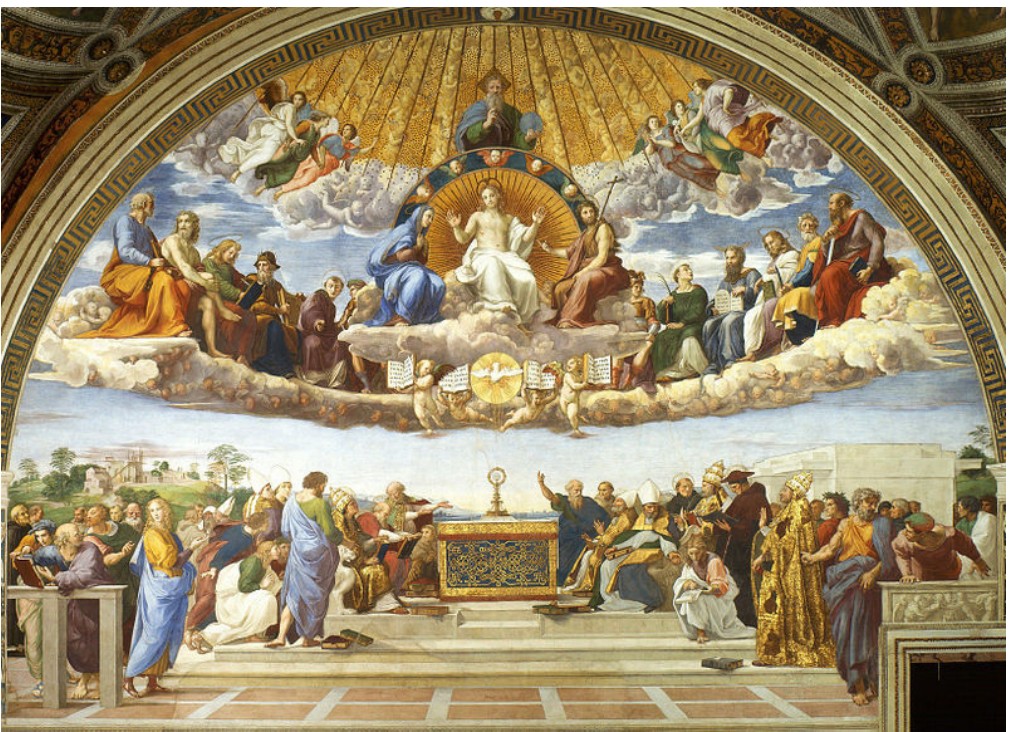

**Figure 10.** Raphael, *Disputation of the Holy Sacrament*, 1509. Fresco, 5 m × 7.70 m. Stanza della Segnatura, Palazzi Pontifici, Vatican.

## 5. Roundels and Painted Medallions

The last possible reason behind the preference for circular compositions has nothing to do with a meaning or idea, but with an artistic genre and with shape, for they might have originated from Roman clipeus, medallions, and coins, among other objects bearing images. As the Spanish medievalist art historian Manuel Castiñeiras states, 'In the Middle Ages there existed a close relation between artistic models and numismatic types, establishing a fruitful dialogue between coining, 'sumptuary' art and monumental art' (Castiñeiras 2006, p. 250).

As exemplified by the historical process involving the clipeus shape, from its use in military shields to its incorporation in the iconography of a figure enclosed in a clipeus, the prevalence of circular objects for sharing ideas, for imposing a hierarchical order, or as things of value might have given rise to circular compositions and iconographic types. The frequent presence of roundels in the illuminated books examined below, which might be explained by their origin in coins and medallions, seems to be connected with these visual references and cultural objects referred to here.

This brings to mind one of the mystical visions of 'Ara Coeli' in which the Sybil prophesized the Coming of Christ, contained in the Limbourg Brothers' *Très Riches Heures* (Musee Condé, ms. 65 fol 22r) (1412–1486), representing the iconographic type of the Virgin and Child above a waxing moon, inscribed in the sun. This is yet another example of the merging of pagan and Christian concepts, so commonplace in these circular compositions, and might also be the case of another image appearing in the Master of the Munich Golden Legend's illuminated *Book of Hours* (1430), containing several roundels accompanying the main image. In sum, the model is to be found in many different works.

The accent is now placed on the medieval visual culture of the Iberian Peninsula, an especially important European region in the twelfth and thirteenth centuries, thanks to the Arabic influence and its flourishing culture, whose libraries, housing books on medicine, alchemy, and astrology/astronomy, amazed inquisitive Western and Northern European Christian thinkers, as Davies observes (Davies 2009, p. 25). In the Christian kingdoms of the Iberian Peninsula, following the Roman example as in other realms, coinage and its iconographic significance served as a way of promoting the powers that be. Kings disseminated their image, associating it with displays of power and linking it to the religious realm, for example, by inscribing religious symbols or legends, like Dei gratia rex, on their coins (Estrada-Rius 2006, pp. 241–42).

In some Christian realms, particularly those with Muslim neighbours, coinage served to convey religious messages. This is evidenced by a twelfth-century penny bearing the religious legend dextera domini, meaning the hand of God (MNAC reference 034100-N), discovered in the episcopal see of Vic, in the Catalan county of Barcelona, dating from only a few decades after the death of Abbot Oliba (c. 971–1046), the former bishop of Vic, during whose time the area had experienced a golden age (Figure 11).

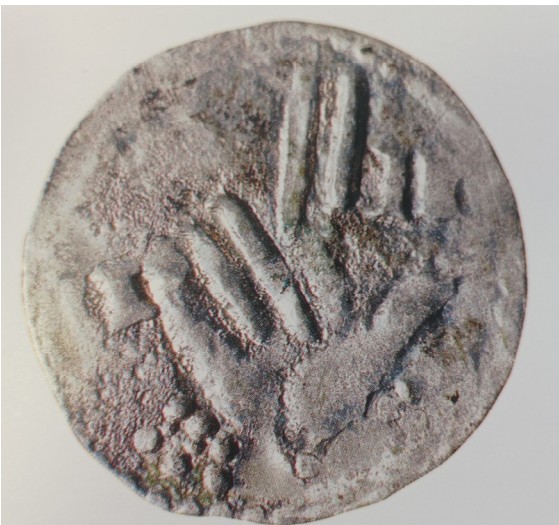

**Figure 11.** Episcopal mint of Vic (Catalonia), *Hand blessing*, second half twelfth century. Silver Coin, 17 mm. MNAC, National Art Museum of Catalonia, 034100-N https://www.museunacional.cat/en/colleccio/diner/bishopric-vic/034100-n. (accessed on 1 October 2023).

Returning to illuminated manuscripts, the *Lapidario*, one of the most remarkable works on astrology, was produced in another of the Christian realms of Iberia, in this case the kingdom of Castile. In Domínguez Rodríguez's opinion, the *Lapidario*, which reflects both Western and Eastern influences, is a syncretic work maybe of Greek origin, specifically from the Hellenistic period, but which would have been transmitted through the Arabic sources (Domínguez Rodríguez 2007, p. 108). This essential work, several copies of which are conserved in Madrid, El Escorial, and the Vatican, is a matchless astrological-talismanic compilation, containing several treatises (Weill-Parot 2002, pp. 123–25). The translations performed at the court of Alfonso X appear to be the only editions of the *Lapidario*, illustrated with illuminated miniatures, to have survived in the West (González Sánchez 2015, p. 153).

The purpose of the *Lapidario*, probably the most sumptuous work of its kind produced in Europe during the thirteenth century, was to extol a king who was a veritable patron of the arts. Immensely wealthy thanks to the lands conquered by his father Ferdinand III (1199–1252), called the Saint, in the valley of the river Guadalquivir, including the major cities of Jaen and Seville, he lavished money on the humanities, among other things.

The artworks produced at the court of Alfonso X reflected syncretic astrological theories strongly influenced by the Neoplatonic and Ptolemaic currents. The *Lapidario*,

which is no exception, is philosophically grounded in Neoplatonic notions, such as the celestial world of the macrocosm reflected in the microcosm of the sublunary world and its stones. To this should be added the idea that all living beings, ordered by their level of perfection on a ladder, are linked by a current of universal sympathy, with God at the root of everything, in a system comprising cosmic bodies and eight or nine spheres, depending on the source.

More than a book on geology, however, the *Lapidario* is a vade mecum, a pharmacopeia, a remedy book (Domínguez Rodríguez 2007, p. 177). In one section, the author draws parallels between 360 stones and the 360 degrees of the zodiac, and in another, he links 36 stones to the decans. The majority view in academia tends to hold that Alfonso X not only promoted research, but also chose the books to be translated, supervising final versions and writing styles (González Sánchez 2015, pp. 18–25).

The pages of the chapter devoted to each constellation are divided into two columns with its name appearing at the top of each double page. The stones are highlighted in red ink, whereas the rest of text is written in black ink. The text descriptions are accompanied by illuminated miniatures visually explaining how to extract the stones, to which should be added other miniatures indicating the sign or constellation in which this should be done.

Although the text descriptions are translations, the images illustrating them correspond to the originals (García Avilés 2011, pp. 108–9). Domínguez Rodríguez singles out Capricorn (79r), which, along with the rest of the constellations, is inscribed in a medallion, as the best drawing in the entire manuscript (Figure 12) (Domínguez Rodríguez 2007, p. 127). The text description of each stone, below the drawing representing it, starts with an illuminated capital, with each page generally containing two illuminated capitals and two representations of the constellation and sign of the zodiac in question, also inscribed in a medallion, thus forming a contrast between the rectangular text columns and the circular images.

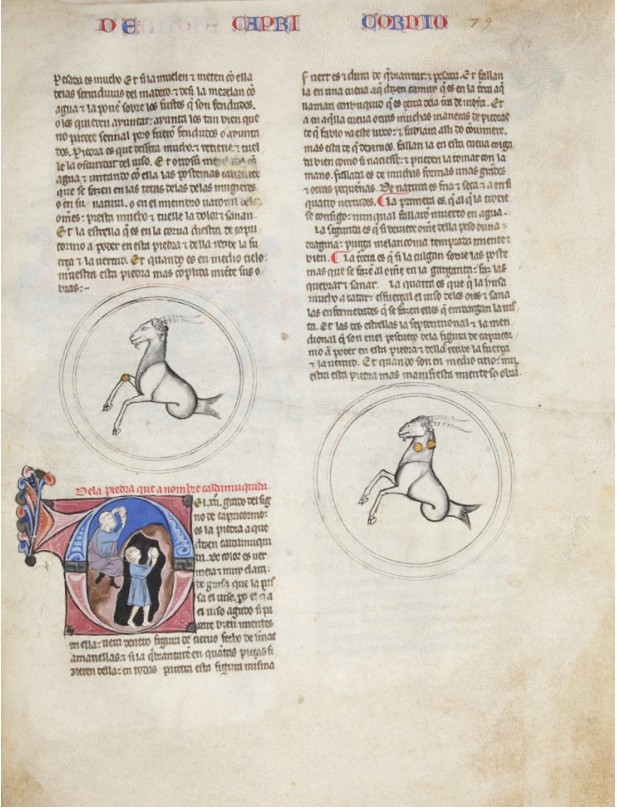

**Figure 12.** Alfonso X, *Lapidario*, c. 1260. Illuminated manuscript (parchment), 40 cm × 29 cm. El Escorial, RBME h-I-15, f. 79r https://rbme.patrimonionacional.es/s/rbme/item/13127#?c=&m=&s=&cv=171&xywh=-97,342,1197,674 (accessed on 1 October 2023).

Several books on astrology were produced in the scriptorium of the court of Alfonso X, most of which were translations of Arabic works[47]. A manuscript housed in the Spanish National Library, commencing with the *Libro de las figuras de las estrellas fijas de la octava esfera* (sixteenth century), also includes the *Lapidario*, which begins with a section in which the constellations are visually represented in wheels or circular diagrams (Figure 13). The book's sumptuousness is exemplified by the medallion representing the constellation of Pegasus on one of its pages, an illustration of a better quality than the rest, according to the researcher specializing in the Alfonsine scriptorium Ana Domínguez Rodríguez (Domínguez Rodríguez 2007, p. 99). Although the book was originally produced in the Alfonsine scriptorium (thirteenth century), it was illuminated in the sixteenth century.

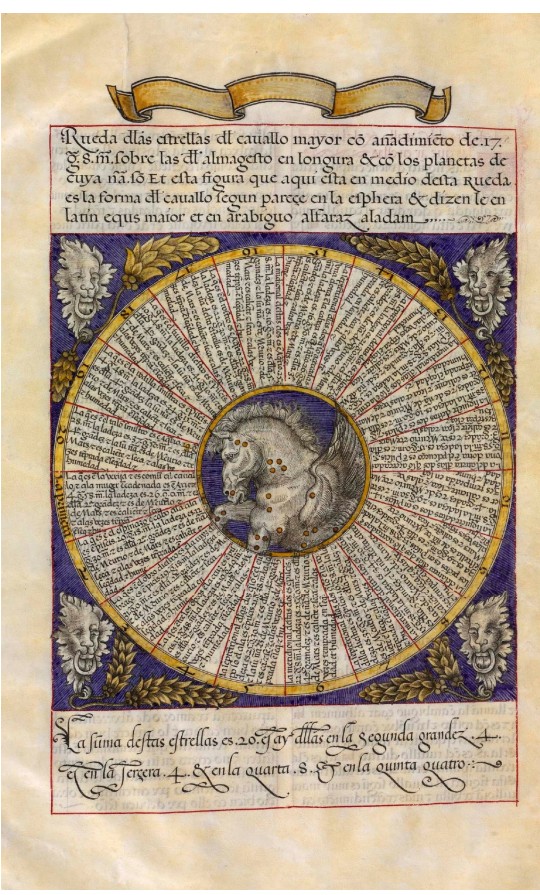

**Figure 13.** Alfonso X, 'Libro de las figuras de las estrellas fijas de la octava esfera,' in *Tratados de Alfonso X sobre astrología y sobre las propiedades de las piedras*, sixteenth century. Illuminated manuscript (parchment), 42 cm × 28 cm. Biblioteca Nacional España mss/1197, tenth image http://bdh-rd.bne. es/viewer.vm?id=0000009637&page=1 (accessed on 1 October 2023).

Yet, the choice of these medallions seems to have had more to do with a lore of visual references, based on Arabic and Roman models and on their own cultural objects such as coins, than with an expression of the universe in which there was a symbolic preference for circular compositions for representing ideas or an analogous way of thinking in which something in the physical world mirrored the perfection of God.

## 6. Conclusions

> Art does not reproduce the visible
> but makes visible
> *Creative Credo*, Paul Klee (1920) (Klee 1961, p. 76)

In the preceding pages, four possible reasons behind the medieval predilection for circular compositions have been examined. Firstly, there was the notion that the universe was circular. Additionally, the circle also served to reflect the form of nature, as a mirror image of God reflecting its beauty and perfection. Thirdly, there were several psychological and symbolic reasons, such as the need to evoke a sense of expansion and security. Lastly, circular compositions might have originated from significant objects, such as coins.

Although it is possible to consider more reasons, for instance, liturgical necessities, these are the four main ones. The article has shown how the influence of writing sources from Platonic, Neoplatonic, Hermetic, and Christian mystic sources helped to establish those reasons. Those texts and ideas contributed to consolidating rounded compositions. Drawing on this background, exemplified by figures like Nicholas of Cusa, Marsilio Ficino, and Giordano Bruno, the essay has illustrated the interconnections between texts and images in visually representing key ideas pivotal for European societies, particularly from the twelfth to the seventeenth century. This perspective illuminates existing continuities.

Linda Safran's words, the 'habit of thinking with circular patterns' (Safran 2022, p. 99) helped create a notion of imago mundi, based on the idea that the world created by God was as perfect as He was. It is noteworthy to point out that, as shown in the preceding pages, to forge those artistic compositions, scientific, philosophical, and theological reasons came together, interwoven in those artistic designs.

At any rate, far from disappearing without a trace, a similar idea of a geometric shape embodying a worldview in an image originated from Copernicus and, subsequently, from Kepler and René Descartes, but with the circle and the circular motion of the spheres being replaced by the ellipse and the spiral.

This idea has left its mark on culture and can still be found here and there. Even in contemporary times, we find artists who exhibit the same attraction to a round arrangement, conveying a sense of marvel at the sight of the cosmos and sharing a feeling of harmony, thanks to the circular composition. For instance, this concept appears to manifest in a photograph taken by Anton Jankovoy in the Annapurna region (Figure 14). After leaving the diaphragm of his camera open for almost two hours, the movement of all the stars was visible, except for that of the Pole Star, whose position is apparently fixed because it coincides with the North Pole.

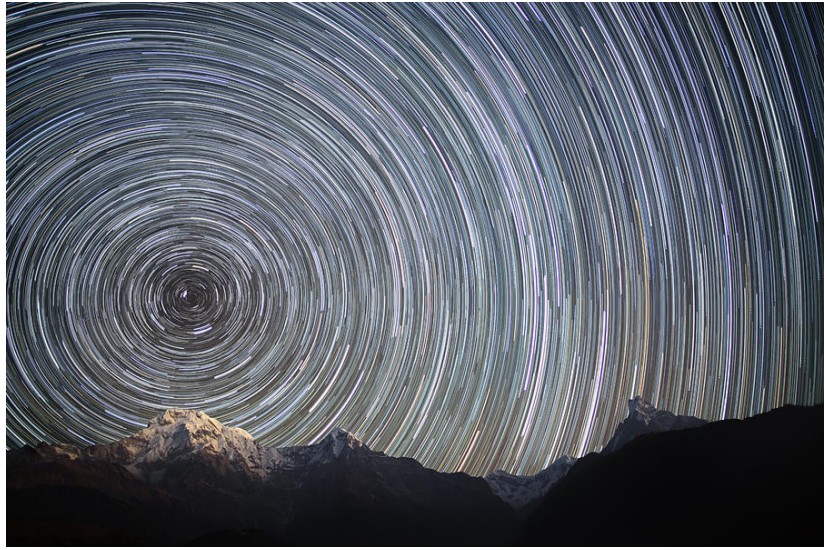

**Figure 14.** Anton Jankovoy, *Head Spinning Himalayas*, 2015. Photography.

**Funding:** This research was funded by NEXTGENERATIONEU and the Spanish Ministry of Universities, grant REQ2021.

**Data Availability Statement:** No new data were created or analyzed in this study. Data sharing is not applicable to this article.

**Conflicts of Interest:** The author declares no conflict of interest.

## Notes

1. As a vade mecum, the treatises compiled in the *Lapidary* serve as a reference book providing instructions on using gemstones to create talismans. These treatises establish connections between the properties of stones, the influence of the stars, and the medicinal implications of utilising them for healing. According to the original prologue, the book was initially written in Arabic. King Alfonso X the Learned acquired it and commissioned its translation into Castilian around 1250. To delve deeper into the *Lapidary*, refer to Section 5 for additional information on this book.

2. Plato, *Timaeus* 33a–c, *Complete Works* (Plato 1997b, p. 1238).

3. Plato, *Timaeus* 38–39d, pp. 1241–43.

4. Collected in (Pedoe 1983, p. 170).

5. Plato, *Timaeus* 33b, p. 1238. At the end of *The Republic* (616–621), Plato mentions the same circular motion, in this case to explain the cosmic movement of souls, which he compares to a spindle: Plato, Rep. X, 616–621 (Plato 1997a, pp. 1219–23)

6. (David 2018, p. 321). Dällenbach mistakenly credits Dante with this famous sentence about God as an infinite sphere, for the idea can already be found in previous Hermetic works, such as the *Book of the Twenty-Four Philosophers*, which, in all likelihood, influenced many other authors, primarily the Florentine poet and writer (Dällenbach 1989, p. 104). Dällenbach should not be held to blame because he is quoting another author.

7. For instance, in *De docta ignorantia*: 'But those who considered the most actual existence of God affirmed that He is an infinite sphere, as it were. I will show that all of these [men] have rightly conceived', in *De docta ignorantia*, I, 12 (Nicholas of Cusa 1990, p. 20). Besides, Cusa maintained that God was an infinite sphere that reached all: see Nicholas of Cusa, *De docta ignorantia*, I, 12, p. 20. As for Bruno, the idea appears in his *De la causa, principio et uno*, V (Bruno 2004, p. 89).

8. *Corpus hermeticum II*, 6–7 (1992, pp. 9–10). The heavenly spheres make their circular motions or kyklophoria (Toussaint 2002, p. 314).

9. Robert Grosseteste was a Franciscan scholastic philosopher from England, Bishop of Lincoln, and well-versed in science. He is primarily recognized for his significant contributions to the study of light and played an important role in the development of the scientific method.

10. In Valentin Weigel's *Philosophia Mystica*, a Rosicrucian book written several centuries afterwards, it is explained that God the Father traces a circle with a compass, the Son becomes the centre of that circle and the Holy Spirit, its circumference (see Arola 2015, p. 190). Later on, it was Masonry that gave importance to the compass for tracing circles and for alluding to sacred geometry and to the Great Architect of the universe. It was also visually used by William Blake, an artist influenced by Hermetism, whose *Ancient of Days* (1794) offered a new version of this figure of God the Creator, this time going beyond His circular perfection. Rosa Alcoy, a researcher specializing in medieval art, links this painting to the medieval *Bible moralisées* (Alcoy 2021, pp. 238–39).

11. In reference to a study on medieval cosmology and the nature of spheres, please see (Edward Grant 1987, pp. 152–73).

12. In an Alfonsine text, the zodiac is defined as the place that is willing to give a soul all the things that are willing to receive it (Domínguez Rodríguez 2007, p. 104).

13. Circular zodiacs were also very popular in Eastern Christianism. According to Linda Safran, 'Zodiac imagery presented in a circular format was widespread in the early Byzantine eastern Mediterranean' (Safran 2022, p. 95).

14. Another circular diagram used for topics relating to astrology was the volvelle, a device designed to locate the position of the moon in the zodiac for medical purposes. In this respect, see, above all, (Page 2002, p. 54).

15. *De radiis* VI, 14 (Al-Kindi 2003, p. 47). 'Some men, thanks to their understanding of the arrangement of the heavens, understood many things hidden in the world of the elements, exploring the secrets of both higher and lower nature' [Own translation].

16. As to the soul as a sphere there is a Gnostic treatise in which it is described as having this shape: (1997, p. 335). For his part, Cusa, following Plato (see n. 4, with the reference to Chapter 10 of *The Republic* and the circular movement of souls), considered that the movement of the Soul of the World, which moved itself and all other things, was circular. This had to be so because its movement was perpetual and contained all motion and beings: *De ludo globi*, I, 40 (Nicholas of Cusa 2000, p. 1201).

17. In the Christian theory of images of the first centuries of the Common Era, they were justified on the grounds that they derived from a Platonic prototype, according to which there was an original image of all things in the divine world and reproductions of them in the earthly world and in art in the form of images, icons, and so forth (Belting 1994, pp. 153–54).

18. De vita triplici III, 19 (Ficino 1980, p. 153).

19. Marsilio Ficino, *De vita triplici* III, 19, pp. 153–54.

[20] Nicholas of Cusa, *De ludo globi*, 1, 11–13, pp. 1186–87.

[21] In his *The Vision of God* (1453), Cusa considers theology as circular, for the attributes of God affirm each other in a circular relationship. Moreover, in these attributes, all otherness is unity and all diversity, identity (they are different but, at the same time, one and, therefore, diverse and simultaneously one and the same). See *De Visione Dei*, III, IX (Nicholas of Cusa 1988, p. 684).

[22] Plato, *Timaeus* 92c, p. 1291.

[23] Nicholas of Cusa, Docta ignorantia I, 11, 30, p. 18.

[24] Secondary sources: (Yates 1964, p. 247; Tessicini 2016, p. 138). Bruno as a disciple of Nicholas of Cusa: (Conty 2012, p. 465).

[25] Lampas triginta statuarum, 64, XXVI (Bruno 2003, p. 1023).

[26] *De imaginum, signorum et idearum compositione* I, 1, 2 (Bruno 1991, p. 10).

[27] Giordano Bruno, *De imaginum, signorum et idearum compositione* I, 1, 13, p. 28

[28] Ibid. I, 1, 14, p. 40.

[29] Ibid. I, 1, 17, p. 44.

[30] Ibid. I, 1, 1, p. 7.

[31] *De gli heroici furori* II, II (Bruno 2013, pp. 281–87).

[32] In (Yates 1964, pp. 393–94).

[33] The philosopher talked about five degrees of all things, from the body, plus its attributes, soul, and angelic nature, to God, in five degrees in hierarchical order: see Theologia platonica, I, I (Ficino 2001, p. 7).

[34] Giordano Bruno, *De la causa, principio et uno*, V, p. 93. Secondary source: (Yates 1964, p. 248).

[35] In the introduction to the Spanish translation of Luca Pacioli's *De divina proportione*, it is contended that a work of art ought to reflect the universe '*allo specchio*': (González 1991, p. 25).

[36] Nicholas of Cusa, *De docta ignorantia* I, XII, 33, p. 20. Cusa also identifies God with an infinite line, which then becomes a triangle (the Trinity), a circle for unity, and a sphere that unites opposites: Ibid. I, X, 27, p. 16. This interpretation has been understood, in the first case, as a reference to exact mathematics, in the second, to speculative mathematics, and, in the third, to mystical mathematics (Machetta and d'Amico 2003, p. 162).

[37] Although, as Durand asserts, the golden ratio would be spiral in shape (Durand 1999, p. 303).

[38] Pacioli was able to develop his intellect thanks to the splendid library of Federico da Montefeltro (1422–1482), to which he had access, probably through his teacher Piero della Francesca. Pacioli collaborated with Leonardo da Vinci when both were living in Milan, hosted by Ludovico Sforza (1452–1508). Leonardo even illustrated *De divina proportione*.

[39] Luca Pacioli, *De divina proportione* XXXI, pp. 67–73.

[40] This symbolism is to be found in alchemy, in which a circle with a focal point signifies the Sun. In his *Psychology and Alchemy*, C. G. Jung holds that this figure signifies protection: *Psychologie und Alchemie*, 63 (Jung 1980, p. 87).

[41] In Book II of *De occulta philosophia* (1533), a reference work in this regard, H. C. Agrippa explains some of the uses of the circle in these practices, considering it as the most suitable figure for bindings and summoning spirits—depending on the place—owing to its infinite nature. As with other Hermetic thinkers, for him, this geometric shape is perfect: *De occulta philosophia* II, 23 (Agrippa 1995, p. 330). This surely explains why in the history of the occult and magic there is a tradition of representing magicians conjuring up spirits in a magic circle. One of the most common Western images of magic is an extravagantly dressed and hatted character drawing a magic circle (the Solomonic circle) on the ground, surrounded by letters and strange signs, before invoking spirits or demons. The same circle can even be found in *Les véritables clavicules de Salomon*, a grimoire written as late as in the eighteenth century. But it is John William Waterhouse's *The Magic Circle* (1886), that is a shining example of magical imaginary.

[42] In C. G. Jung, *Psychologie und Alchemie*. The bottom line here is whether the mind perceives it or not. Those variations explain why this structure is only perceived by a few: C. G. Jung, *Psychologie und Alchemie*, 329–330, pp. 240–41.

[43] *The Enneads*, VI 9, 8 (Plotinus 2016, pp. 892–93). The journey that a mandala offers beholders is from its circumference to its centre, where its main deity reigns (Leidy 1997, p. 41). According to Macrobius, as the soul descends through the cosmic spheres and each planet, it acquires the qualities that it will possess once it has a body: *Commentary on the Dream of Scipio*, I, 12, 13–14 (Macrobius 1990, pp. 136–37). In his commentary on Plato's *Symposium on Love*, Ficino mostly followed Macrobius's ideas, albeit introducing small modifications: *De amore* VI, IV (Ficino 1987, pp. 117–19).

[44] For further information, see (Ferrer-Ventosa 2023).

[45] Tibetan mandalas integrate both with a symbolic sense of harmonizing them (Arnheim 1982, p. 117), due to their non-dualistic belief.

[46] Nicholas Temple and Cecilia Panti have connected the rose windows of Lincoln Cathedral to Grosseteste's scientific ideas on light. (Temple 2016, pp. 29–58; Panti 2016, p. 74–76).

[47] Work in the scriptorium was conducted in several steps: *transladar* or translating; *capitular* or dividing the text into chapters; *emendar* or correcting; *endreçar* or style editing; *glosar*, namely, when a specialist was called upon to assess the results or even to

elaborate on them; there was also sometimes an additional step that could be called *ayuntar*, which involved selecting sources and compiling them coherently (González Sánchez 2015, p. 168).

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
