# Peer review of "On the Perfect Sphere: The Preference for Circular Compositions for Depicting the Universe in Medieval and Early Modern Art"

_religions, doi:10.3390/rel15020171_

Round 1

Reviewer 1 Report

Comments and Suggestions for Authors

General comments:

The article aims to explain why ‘cultures’ of the Middle Ages and the early modern periods frequently favoured circular compositions in visual representation of the cosmos. It divides answers into four: 1. the cosmos was understood as circular, 2. symbolic and psychological appeal of circular compositions, 3. a connection between nature and the divine understood to be reflected in the circles, 4. influence of objects, predominantly coins. The article tries to bring together study of ideas, philosophy, nature and religions with artistic representation and this is convincing and a contribution to several fields. 

The discussion is rich and includes diverse textual and visual evidence. The author draws upon diverse philosophies spanning Christianity, Islam, Buddhism, Ancient Greek schools of thought and more. This is a strength of the paper, along with the diverse visual examples of  circular representations of the cosmos. Noteworthy is the example in the conclusion, showing a contemporary artistic take on the topic. While it is convincing that the one informed the other there is no attempt to show that the artistic traditions at hand were made by artists and patrons who had access to the knowledge cited. For example, the author writes that 'One of the most compelling reasons behind this penchant for circular compositions has to do with the way in which the ancient Babylonian, Greek, Jewish and Roman cultures conceived the universe as a circular entity, forging a vision that would subsequently give rise to the Christian worldview”. It would follow that to show this influence the author would either cite extant research or explain how the ideas of Babylonian, Greek, Jewish and Roman culture came to the knowledge of the artists and patrons.

To maximise the influence and rigour of the paper it is important to clearly present the state of research: how have scholars treated circular compositions thus far? what types of explanations for their popularity have been given and how do these explanations offered by the author expand on them? Another important issue is the contextualisation of sources: neither the written nor the visual sources are carefully presented within their historical context. the author pivots between different periods and places and does not explain the temporal framework of the investigation. Examples span almost 1500 years. In line 46 the author mentioned 'the images created at the time' but with no indication of when that time may be. Authors names are given without dates (for example Luca Pacioli, line 57 or Giordano Bruno, line 71). General statements would benefit from being backed by examples, for example line 70: ‘God, as the epitome of perfection, was often conceptualized or visualized in a rounded form’, examples should be given in a footnote. There is some generalisation in the discussion, treating medieval and early modern artists as one group and not mentioning regional differences (for example Byzantine art). 

‘Culture’ is not defined here as relating to a specific time and place, leaving conclusions so generalising as to loose their weight. Indeed the broad references are both a strength of the paper and a potential methodological problem at one and the same time. On the one hand the author has compiled an impressive set of sources, from Ancient Egyptian concepts of the universe, Greek philosophy and medieval medicine, all of which inform the authors questions directly. Therefore there really does emerge a picture of ‘culture’ along different periods and different places using the circular diagram as a conceptual understanding of the world. At the same time there is a lack of historical focus: the author rarely gives context to the works of the authors cited, neither stating where and when they were active or the extent of their influence beyond their place and times. 

Many terms are not explained in the article, for example Hermetic and Neoplatonic cultural horizons. These are important for following the paper and should be explained upon first use. There are several art historical investigations of circular representations, notably Bianca Kühnel’s book The End of Time in the Order of Things and an edited volume The Diagram as Paradigm: Cross-Cultural Approaches edited by J. F. Hamburger, D. A. Roxburgh and Linda Safran or The Diagrammatic Mode in Byzantium.

The conclusions, examples and arguments are all compelling and interesting.

Specific comments:

- Figure 4 appears to be flipped: the Latin is reverse. Also the DOI link leads to a similar but not identical image. 

  • The whole explanation about the perception of the universe (lines 181-186) needs references. 
  • Lines 327-332 need to be referenced. 
  • There is inconsistent reference to the images and inconsistence in properly presenting the images. 
  • Specialised terms should be explained as they make certain sentences very hard to read for non specialists, for example: 'Regarded as a vade mecum or pharmacopeia, the Lapidario is devoted to astrological 90 and talismanic themes combining Western and Eastern influences’ (lines 90-91). 
  • Literary works, for example the Lapidario, should be presented at least in a footnote to explain genre, author, dates, schools of thought etc. 
  • The same regards thinkers and authors, for example line 156-7: “Christian thinkers also pondered on this subject. For his part, the medieval scientist Grosseteste argued…” in order to correctly understand the arguments the reader needs life years in brackets and a footnote on who this is.  

Author Response

My response is in the attached file. Thank you very much for your comments and suggestions. 

Reviewer 2 Report

Comments and Suggestions for Authors

please read the comment.

Comments on the Quality of English Language

please read the comment.

Author Response

(The authors gave the same response as above.)

Reviewer 3 Report

Comments and Suggestions for Authors

I am honored to have had the opportunity to review this paper and am impressed with the depth of research the author had undertaken, however, for this paper to have the desired impact on scholarship that I am sure this author desires I would suggest some revision and a narrower focus. At this time the thesis as stated -- to explain medieval and early modern artists preference for circular compositions -- is too broad. I found many of the sections compelling, especially the "Shape of the Universe," but others, such as "Circular compositions and the symbolism and psychology deriving from the forms depicted" were too broad and the evidence felt disconnected. The symbolism section seemed the most broad and when reading that section I found myself thinking about historical contexts that might better explain the formal qualities of a work of art. This was especially true for Hagia Sophia, which is a complicated example and the shape is more readily explained by other influences or contexts, such as the liturgical or even building that offered historical models. I think there might be a building with less complex histories that could be used here, but this whole section felt the least contextualized to me. I found the parts of the author's argument that was tied to textual sources to be much richer and more convincing. 

Therefore, I would encourage the author to rethink the essay along the lines of a more focused thesis, one that is more closely connected to the revival of sources across time, which I felt to be a very successful part of the argument. 

I was intrigued by the notion that the author would study the "the break that [circular compositions] created with their rectangular settings" but I am not sure that section bears fruit and could be cut. If the author wants to keep that they might look more at Leonardo da Vinici's Vitruvian Man as well as the text by Vitruvius from which it derives. Renaissance architectural historians have done excellent work with the influence of the cirlce and the square on the change from what we can medieval to what we call Renaissance building and structures like Bramante's Tempietto and the many designs for the New St Peters bear this out (which is cited here, but the braccia (arms) of the courtyard and not the dome, which is where we might see a better example of this idea. 

I also think the author might want to limit the number of works of art referenced. Some are included as links, others as images inserted into the text; in some cases the authors talks about the images directly, which I would always encourage, while in a few cases, such as the Girona Tapestry of Creation, there is little discussion.  

Comments on the Quality of English Language

The quality of the English in the opening paragraphs could use some revision, but the arguments in the essay are well articulated (in terms of the quality of the English) 

Author Response

(The authors gave the same response as above.)

Round 2

Reviewer 2 Report

Comments and Suggestions for Authors

Please read the comment.

Comments on the Quality of English Language

Please read the comment.

Author Response

Many thanks for your insightful suggestions and your work

Reviewer 3 Report

Comments and Suggestions for Authors

Overarching suggestions:

1) The essay is still doing a lot. I remain convinced that a narrower focus would have a greater effect on the reader. It is less clear to me what exactly the author is adding by doing so much. The more significant contributions seem to be in the second and fourth sections and parts of the fifth section. 

2) I remain concerned that the author does not discuss some of the Figures illustrated in the essay. For example, it is unclear to me why Figure 2 is included. The Catalan Atlas is not mentioned in the body of the essay. It is the same for Figure 3. The same is true for Figure 4 (Nuremberg Chronicle). I can see how they might be related to the section but the author does not discuss these images specifically, they seem to be being used as general illustrations of a theme and that is not appropriate for an art historical analysis. In fact, for Figure 3 the author noted a second image by name (Pere Serra’s Retable of the Holy Spirit), which is connected via a hyperlink rather than discussing the Codex Vindobonensis. 

3) When the author does discuss the Figures, they should note that in the body of the text. For example, on page 7, line 251 the author discusses the Revised Aratus latinus, they should follow this identification with (Figure 5), which illustrates that text. This pattern of discussion and identification should be followed for all of the figures included. Additionally, the caption for Figure 5 is missing the title Revised Aratus latinus.

5) In my original response, I had also indicated that I was unclear as to why some images were hyperlinked and others were included as Figures. I might suggest only including Figures of works of art you discuss, thereby including Figures for the hyperlinks and omitting anything you do not discuss in the body of the text. 

6) Section 5 does not stand as well on its own -- consider combining it with Section 3. You might also consider moving section 4 BEFORE section 3. Then the two formal sections would be together and the two philosophical sections would be together. Another option, move the section that talks about the clipeus to the beginning of Section 3 where you discuss formats such as the tondo. Then reconnect the section on the Lapadario to the part of the essay where you discuss this the first time. This would better connect this section to where you discuss the shape of the universe. 

7) The conclusion needs to be revised. There is some repetition as a result of the newly added sections and I am not sure it offers the clarity in closing out a complex argument that the author would like. Some of the revisions seem abrupt in this section and need to be better integrated. 

Corrections by page

p. 1 line, 5: I would remove "insightful" from the abstract. That term sounds like something an external reader would say about the work but not something an author would say about their work. 

p. 1, line 12: I would also change "visual arts symbolism" to symbolism in the visual arts 

 p. 1 lines 26-31: Suggest revising to: How does a culture envision something as inherently ineffable, or at least beyond the human capacity for visualization, as the entire universe and beyond? This essay will delve into an aspect of medieval aesthetics that linked religious, philosophical, and scientific ideas about the shape of the universe as well as ways of relating to it and representing it using circular compositions.  

p. 1 lines 31-32, omit and combine with the next two sentences. 

p. 3, combine two sentences to avoid repetition (lines 68-71)

p 3.  79 - integrate the authors in parenthesis as you did for the others. Same for line 81 -- integrate as in lines 77-78. Apply this to the whole paragraph. 

pg. 3, line 87 - period after point, begin new sentence.

p. 4, line 133 -- it is unclear what the added part is modifying - pharmacopeia or Lapadario.

p. 8, line 283, you say the aforementioned Dante but I think this is the first mention. 

p. 9, line 334, omit "Anyway"

p. 10, lines 345-347 - while I appreciate you changed "break" to "tension", this is still an unfinished argument and it is very intriguing. I would either expand it or omit it. 

p. 10, line 358, include (Figure 6) after this first sentence. 

p. 11, lines 383-384 change to: "The golden ration was applied to both the floor plan and the elevation drawing of Florence Duomo."  It's unclear to me what you mean by "as it could not have been done otherwise" so I would omit this. 

p. 12, lines 442 and 444 - please see my note about hyperlinks and/or images you do not discuss in the body of the paper. I'd also argue that a fuller discussion of imago mundi would be a welcome addition to this essay. 

p. 13, lines 464-466 -- as I noted in my previous comments I think the author should engage architectural history here, especially discussions of Bramante's Tempietto and the theories of the circle and square associated with the shifts from medieval basilica plans to centrally planned Renaissance churches. I also remain unconvinced that the Braccia of Bernini for St. Peter's is the best example here, there might be a better connection for this example if the author looked at the scholarship on the planning drawings by Bernini for the Braccia. 

p. 14, lines 470ff - there is work that connects Grosseteste's writings to the Rose windows at Lincoln that would be relevant here. 

p. 14 lines, 490, this example feels too brief to be meaningful here. 

p. 15: The Raphael example is still disconnected from its larger context. Given the School of Athens includes philosophy and even an image of Renaissance architect Bramante as Euclid using a compass, the broader context seems relevant. 

Comments on the Quality of English Language

The introduction and the conclusion could use revision. The body of the essay needs only minor attention but an external editor who could revise for clarity would be helpful. 

Author Response

(The authors gave the same response as above.)
